# Correlation of Clinical Parameters with Intracranial Outcome in Non-Small Cell Lung Cancer Patients with Brain Metastases Treated with Pd-1/Pd-L1 Inhibitors as Monotherapy

**DOI:** 10.3390/cancers13071562

**Published:** 2021-03-29

**Authors:** Konstantinos Rounis, Marcus Skribek, Dimitrios Makrakis, Luigi De Petris, Sofia Agelaki, Simon Ekman, Georgios Tsakonas

**Affiliations:** 1Thoracic Oncology Center, Theme Cancer, Karolinska University Hospital, 17164 Stockholm, Sweden; marcus.skribek@ki.se (M.S.); luigi.depetris@ki.se (L.D.P.); simon.ekman@ki.se (S.E.); georgios.tsakonas@ki.se (G.T.); 2Department of Oncology-Pathology, Karolinska Institutet, 17177 Stockholm, Sweden; 3Department of Medical Oncology, University General Hospital, 71110 Heraklion, Crete, Greece; dmakrak@uw.edu (D.M.); agelaki@uoc.gr (S.A.); 4Division of Oncology, University of Washington Medical School, Seattle, WA 98195, USA

**Keywords:** NSCLC, brain metastases, immunotherapy, intracranial outcome

## Abstract

**Simple Summary:**

We analyzed data from patients with advanced Non-Small Lung Cancer (NSCLC) and brain metastases (BM) who were treated with PD-1/PD-L1 inhibitors as monotherapy at Karolinska University Hospital, Sweden, and University Hospital of Heraklion, Greece in order to identify parameters that can potentially affect the intracranial (IC) outcome of these individuals. We assessed IC immunotherapy (I-O) efficacy in the patients who had BM prior to I-O administration, radiological evaluation for IC response assessment and they had not received any local CNS treatment modality for ≥3 months before I-O initiation. Age < 70 years old, prior radiation treatment to CNS, and primary (BM present at diagnosis) BM were associated, at a statistically significant level, with an increased probability of achieving IC disease control in our cohort. These results suggest that specific clinical parameters may potentially influence IC outcomes in NSCLC patients with BM.

**Abstract:**

There is a paucity of biomarkers for the prediction of intracranial (IC) outcome in immune checkpoint inhibitor (ICI)-treated non-small cell lung cancer (NSCLC) patients (pts) with brain metastases (BM). We identified 280 NSCLC pts treated with ICIs at Karolinska University Hospital, Sweden, and University Hospital of Heraklion, Greece. The inclusion criteria for response assessment were brain metastases (BM) prior to ICI administration, radiological evaluation with CT or MRI for IC response assessment, PD-1/PD-L1 inhibitors as monotherapy, and no local central nervous system (CNS) treatment modalities for ≥3 months before ICI initiation. In the IC response analysis, 33 pts were included. Non-primary (BM not present at diagnosis) BM, odds ratio (OR): 13.33 (95% CI: 1.424–124.880, *p* = 0.023); no previous brain radiation therapy (RT), OR: 5.49 (95% CI: 1.210–25.000, *p* = 0.027); and age ≥70 years, OR: 6.19 (95% CI: 1.27–30.170, *p* = 0.024) were associated with increased probability of IC disease progression. Two prognostic groups (immunotherapy (I-O) CNS score) were created based on the abovementioned parameters. The I-O CNS poor prognostic group B exhibited a higher probability for IC disease progression, OR: 27.50 (95% CI: 2.88–262.34, *p* = 0.004). Age, CNS radiotherapy before the start of ICI treatment, and primary brain metastatic disease can potentially affect the IC outcome of NSCLC pts with BM.

## 1. Introduction

The central nervous system (CNS) is a frequently metastasized site in non-small cell lung cancer (NSCLC) [1,2,3]. Brain metastases (BM) constitute an adverse prognostic factor for NSCLC [4,5,6] and despite the improvements made in diagnosis, staging, and treatment for NSCLC patients with BM and non-oncogenic driven tumors, prognosis remains poor with an average overall survival ranging from 5 to 9 months [7,8].

Immunotherapy (I-O) with immune checkpoint inhibitors (ICIs) has transformed the treatment landscape of NSCLC, offering a tool in the oncologists’ armamentarium for the achievement of durable remissions in a substantial proportion of patients [9,10,11]. However, patients with BM have been underrepresented in the pivotal clinical trials that lead to the regulatory approval of ICIs for NSCLC, since only patients with stable BM were included [12,13,14,15,16,17,18,19,20,21,22]. In addition, the application of radiation treatment or surgical excision of BM was allowed prior to the initiation of ICIs in these trials, rendering the evaluation of intracranial (IC) efficacy of ICIs not possible.

Goldberg et al. [23] in a single cohort phase II trial examined pembrolizumab activity in active BM due to NSCLC and reported an IC objective response rate (ORR) of 29.7% in patients with PD-L1 expression ≥1% in their extracranial lesions. However, in the aforementioned study, the presence of neurological symptoms due to BM and corticosteroid requirements constituted exclusion criteria [23]. A retrospective study by Hendriks et al. [24] analyzed 255 patients with BM due to NSCLC treated with ICIs and 73 out of these 255 patients were categorized as having active BM. In their analysis, they reported an IC ORR of 27.3% in the subgroup of patients with active BM. In a similar retrospective study, our research group demonstrated an IC ORR of 24% in individuals with advanced NSCLC and active BM, whereas the presence of neurological symptoms due to BM and the administration of steroids did not affect IC response rates [25].

In addition to the limited availability of clinical data, there is a paucity of biomarkers for the prediction of IC outcome in ICI treated NSCLC patients with BM. Extracranial PD-L1 expression is the only available biomarker that has been associated with improved clinical efficacy and survival in prospective trials [12,13,14,15,16,17,18,19,20,21,22]. However, PD-L1 expression levels in extracranial lesions have not been evaluated for the prediction of IC outcome in NSCLC patients with BM treated with ICIs.

We hypothesized that various clinical parameters may potentially affect these patients’ IC outcomes after ICI administration. To this end, we conducted a multicenter retrospective study in order to investigate the clinical variables that could potentially affect the intracranial outcome in NSCLC patients with BM treated with ICIs.

## 2. Results

### 2.1. Patient Characteristics

Patient characteristics are summarized in Table 1. The median age was 69 years (range: 40–84). Of the patients, 24 (72.7%) had BM at their diagnosis (primary) and 18 (54.5%) exhibited neurological symptoms due to their BM. The IC response to ICIs in 23 (69.7%) patients had been evaluated comparing their MRI before and after ICI treatment and in the remainder 10 (30.3%) with CT scan. Prior to ICI initiation, 12 (36.4%) had >3 BM and the median size of the largest BM was 17 mm (range: 4–50). Of the patients in our cohort, 25 (75.8%) were categorized as having active BM and 21 (63.6%) had experienced IC disease progression to their previous treatment. None of our patients had confirmed leptomeningeal dissemination and none had continued ICI beyond IC progression. The median IC time to progression (TTP) was 4.3 months (95% CI: 2.81–5.79 months) (Figure 1). Neurological deterioration that was attributed to the BM was experienced by 5 patients earlier than 6 weeks after ICI initiation. These 5 individuals had a subsequent radiological evaluation that confirmed IC disease progression in comparison with their baseline scans and they were classified as having IC disease progression.

### 2.2. Results from Response Assessment

We initially examined the effect of age, number of BM, and the diameter of the largest BM as continuous variables on IC disease control (PR or SD) (DC) rates. Advanced age was associated with reduced IC DC rates (*p* = 0.045) (Figure 2). The diameter of largest BM (*p* = 0.163) (Appendix A) and number of BM (*p* = 0.868) (Appendix A) did not affect IC DC rates.

Afterward, we evaluated the effect of the studied categorical parameters on IC DC rates. The results are depicted in detail in Appendix A. Age < 70 years old (*p* = 0.019), adenocarcinoma histology (*p* = 0.038), previous cranial radiation therapy (RT) (*p* = 0.022) (Figure 3A) and primary BM (present at diagnosis) (*p* = 0.011) (Figure 3B) were correlated with increased rates of IC disease control. In addition, patients with stable BM before ICI treatment start (*p* = 0.001) had superior rates of IC disease control, whereas the presence of neurological symptoms attributed to BMs (*p* = 0.373) and steroid administration >10 mg of prednisolone equivalent (*p* = 0.221) did not affect IC control rates.

Finally, we calculated the odds ratio (OR) of each studied covariate on the probability of developing IC PD. The OR of the parameters histology (adenocarcinoma vs. non-adenocarcinoma) and active vs. stable BM were not calculated because none of the patients with non-adenocarcinoma histology experienced IC disease control, and none of the patients with stable BM experienced IC PD (the denominator was zero in these two parameters). The results are demonstrated in the forest plot in Figure 4A and in Table 2. Non-primary BM, OR: 13.33 (CI: 1.424–124.880, *p* = 0.023), age ≥70 years old, OR: 6.19 (CI: 1.27–30.170, *p* = 0.024) and no previous RT for BM, OR: 5.49 (CI: 1.210–25.000, *p* = 0.027) were the parameters associated with an increased probability of IC PD after ICI administration at a statistically significant level.

### 2.3. Creation of Different Predictive Groups Based on I-O CNS Score

The parameter with the highest OR for developing IC PD was non-primary BM, thus primary BM was allocated 1 point in our scoring system. The second in order parameter with the highest OR was age ≥ 70 years old, thus age < 70 years old was allocated 0.5 points in our scoring system. The third parameter was the lack of administration of previous CNS RT and as such previous CNS radiation treatment was allocated 0.5 points in our scoring system.

Using this point allocation, we created two different predictive groups for IC disease control as a result of ICI administration (Table 3). Group A (good group) consisted of patients with 1.5–2 points on the I-O CNS score and group B (poor group) consisted of patients with 0–1 point on I-O CNS score (Table 4). Patients’ distribution according to their scores on I-O CNS score is depicted in Appendix A. We then examined the effect of the I-O CNS score poor group on the probability of developing IC disease progression after ICI administration. I-O CNS score poor group demonstrated an OR: 27.50 (CI: 2.88–262.34, *p* = 0.004) (Figure 4B). Our scoring system performed with an area under the ROC curve (AUC) = 0.792 (95% CI: 0.631–0.953, *p* = 0.004) (Figure 5).

## 3. Discussion

In this study, we retrospectively analyzed data from patients with NSCLC treated with ICIs from two oncologic institutes, Karolinska University Hospital in Sweden and Heraklion University Hospital in Greece, with the purpose of identifying clinical parameters that affect the IC outcome of these individuals.

Despite the recent advances in the treatment of advanced NSCLC, the management of NSCLC patients with BM remains challenging. In addition, there is a lack of biomarkers for the prediction of IC outcome and proper treatment planning is demanding in brain metastasized NSCLC patients treated with ICIs. Furthermore, PDL1 expression levels have not been evaluated in this setting. The limited clinical data available on the activity of I-O on active BM provided by Goldberg et al. [23] did not include patients with neurological symptoms attributed to their BM or patients with corticosteroid treatment. Similarly, in the retrospective data analysis by Hendriks et al. [24] only 12% of the patients with active BM experienced symptoms due to CNS dissemination. For the purpose of our analysis to be more representative of the actual population pool in the ‘real world setting’, we included patients with active BM and neurological symptoms due to their BM. Moreover, in order to be able to sufficiently assess the effect of I-O on BM, we used a time interval of 3 months prior to the initiation of I-O that the patients were allowed to have received local CNS treatment modalities, and patients who received a combination of PD-1/PD-L1 agents with chemotherapy or CTLA-4 inhibitors were excluded from the final analysis. Finally, the majority of our cohort consisted of patients who had active BM that had progressed on previous therapy.

Patients with BM that were not present at diagnosis but developed later during the course of disease trajectory (non-primary) demonstrated inferior IC responses to I-O. This finding is probably related to the fact that the BM of these patients consist of malignant clones that have emerged under the selective evolutionary pressure of previous systemic therapy and active immunosurveillance and thus have acquired the specific genetic traits for successful immune evasion. Based on these findings, these patients should have a more intense follow-up after ICI administration and the options for the potential application of localized CNS treatment should be exploited since they have a low chance of achieving IC disease control and subsequent clinical benefit of ICI administration. However, due to the small patient sample in our cohort, further validation of this finding is needed in larger patient cohorts.

Advanced age also constituted an adverse predictive factor of IC outcome in our patients. Our data are in accordance with previous retrospective survival analyses, which have demonstrated that age is an adverse prognostic factor in NSCLC patients with BM [4]. In addition, none of the patients in our cohort with non-adenocarcinoma histology experienced IC DC after ICI administration. Non-adenocarcinoma histology has been recognized as an adverse prognostic factor for inferior survival in NSCLS patients with BM [4,26] and as a factor that predicts the increased probability of local recurrence after SRS application [26].

Interestingly, the prior administration of RT was associated with improved rates of IC DC in our cohort. Radiation therapy induces the fragmentation of genomic and mitochondrial DNA that increases DNA sensing in dendritic cells to produce type I interferon for the activation of anti-tumor immunity [27,28]. However, there are limited clinical and molecular data available to serve as a platform for the creation of a model of potential synergy between these two modalities [29]. Nevertheless, despite the heterogeneity of prior radiation treatment modalities in our cohort (SRS and WBRT), the increased rates of IC DC after their application is a promising finding that warrants further investigation in larger patient cohorts.

The clinical score that we developed was able to predict the possibility of disease progression (or disease stabilization as a binary model) as the best response to I-O administration with a high OR and a satisfactory AUC. The large confidence intervals are due to the small number of patients. Due to the small statistical sample, we do not recommend its utilization in daily clinical practice, until these results have been verified in larger patient cohorts. However, our findings are clinically significant and they could potentially guide future research projects.

The major limitation of our study is the small population sample with limited statistical power. The risk for type -1 or -2 statistical errors (especially type -2 which is directly correlated to sample size) is high in small retrospective cohort trials. Selection bias also cannot be avoided. In addition, patients that have received I-O as different lines of treatment were included. Furthermore, patients with no CNS imaging after treatment initiation were excluded, which may result in an underestimation of treatment failures due to early progression or death. Finally, due to the fact that only 6.1% of patients had PD-L1 status < 1% it was not possible to examine the effect of PD-L1 expression levels on the probability of intracranial disease progression. There is a need to validate the proposed I-O scoring system in another larger cohort and it is possible that our scoring system will be slightly altered after it has been validated in such a cohort.

## 4. Methods

### 4.1. Patient Eligibility

We enrolled 33 patients—out of 280 who were screened—with histologically confirmed stage IV NSCLC with BM prior to ICI administration who received standard monotherapy with anti-PD-1/PD-L1 inhibitors as a first or subsequent line of treatment at Karolinska University Hospital, Sweden, and University Hospital of Heraklion, Greece, between December 2016 and February 2020. All treatments were administered according to international guidelines and the treating physician. The study was reviewed and approved by the institutional review boards of the Karolinska University Hospital and the University Hospital of Heraklion and was conducted in accordance with the principles of the Declaration of Helsinki (IDs: 2644, 2020-02636).

### 4.2. Study Design

Our study design is summarized in the flow chart in Figure 6. Patients with *EGFR* mutations or *ALK* translocations were excluded from the analysis before the initial screening.

Our inclusion criteria consisted of the following: baseline BM prior to ICI administration, at least one administration of an anti-PD-1/PD-L1 agent, ICI treatment in the form of monotherapy with PD-1/PD-L1 inhibitors (patients who had received PD-1/PD-L1 inhibitors in combination with chemotherapy or with anti-CTLA-4 antibodies were excluded), a time interval of ≥3 months from the application of a local treatment modality (surgery, SRT/SRS, WBRT) until the initiation of ICIs and a CNS radiological evaluation with a CT or MRI that addressed the effect of ICIs administration on BM (patients who received local CNS treatment modalities after ICI administration were excluded).

The following parameters were analyzed: age, gender, smoking status, Eastern Cooperative Oncology Group-Performance Status (ECOG-PS), tumor histology (adenocarcinoma vs. non-adenocarcinoma), PD-L1 expression levels of extracranial lesions, number of organs with metastatic disease, presence of neurological symptoms attributed to BM, local CNS treatment modality (WBRT, SRS, surgery), line of treatment of ICI administration, presence of active BM, baseline steroid administration with dose equivalent of >10 mg prednisolone for ≥10 days, ICI start/end-date, date of progression and date of death or last follow-up. PD-L1 expression levels were calculated in all 33 patients in the biopsies from extracranial disease sites (lungs, liver, bone, or other). None of the analyzed individuals had PD-L1 levels calculation from brain metastasis. The patients were categorized as having received steroids if they had received >10 mg of prednisolone equivalent for >10 days 15 days before immunotherapy initiation or during the duration of treatment. Intracranial Time to Progression (TTP) was calculated from the initiation of ICI until radiologically confirmed intracranial disease progression. Overall survival (OS) was calculated from the initiation of ICI until death. Response assessment for progressive disease in the CNS was conducted according to RECIST 1.1 criteria [30].

BM were defined as active if they had not been previously treated or they had progressed after prior surgical excision, radiotherapy, or systemic chemotherapy. Primary CNS disease was defined as the presence of BM at diagnosis and non-primary if the patient had developed BM later during his/her disease course but before the initiation of ICI.

### 4.3. Response Assessment

IC response of BM to ICI was assessed according to RECIST 1.1 criteria [30]. Patients were classified as having IC complete response (CRi), IC partial response (PRi), IC stable disease (SDi), and IC progressive disease (PDi) based on their radiological assessment with brain MRI or CT after ICI administration. IC disease control (DCi) was defined as CRi, PRi, or SDi. The IC radiological assessment was conducted by comparing the CTs or MRIs of the patients that were performed up to 12 weeks after their first ICI administration, with their baseline CTs or MRIs before treatment initiation. In addition, for the assessment of IC response, we compared similar imaging methods. If a single individual had radiologically confirmed intracranial progression in a time duration shorter than 12 weeks he was classified as having IC progression.

### 4.4. Statistical Analysis

Descriptive statistics were performed to analyze categorical and continuous variables. We applied the Mann–Whitney U test to examine the effect of continuous nominal variables on their effect on DCi rates. The chi-square test (X2) was used to assess the effect of categorical variables on IC disease control. A univariate binary logistic regression analysis was performed to examine the odds ratio (OR) of each clinical parameter on the probability of developing disease progression as a response to ICI administration. A *p*-value of <0.05 (two-sided test) was considered statistically significant. No multivariate logistic regression analysis was performed due to the low number of patients.

We decided to create a scoring system including the parameters that exhibited the higher adjusted weight on the probability of developing IC DC based on their OR. Three variables would be included, the value with the highest OR would receive 1 point and the other two 0.5 points. Based on this point allocation system two groups would be created, group A (good) with a score 1.5–2 points and group B (poor) with a score 0–1 point.

All statistical analyses were performed using the SPSS 25.00 software.

## 5. Conclusions

In this study, we demonstrate that clinical parameters such as age, CNS RT, and primary CNS metastatic disease can affect the IC outcome of I-O treatment in NSCLC patients with BM and we propose an I-O CNS scoring system for the prediction of intracranial disease control. Further validation of these findings in larger cohorts is required in order to elucidate their potential as predictive biomarkers.

## Figures and Tables

**Figure 1 cancers-13-01562-f001:**
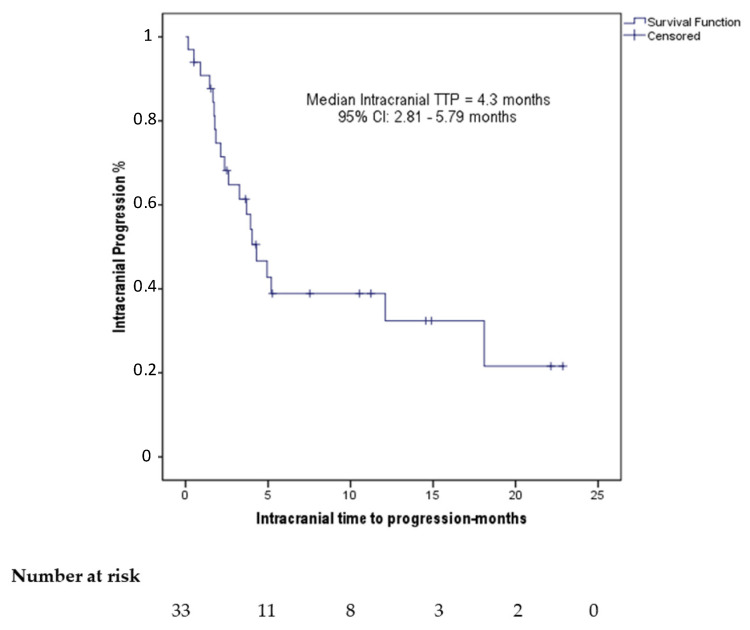
Kaplan–Meier plot demonstrating the intracranial time to progression (TTP) under treatment with ICIs of the patients in our cohort.

**Figure 2 cancers-13-01562-f002:**
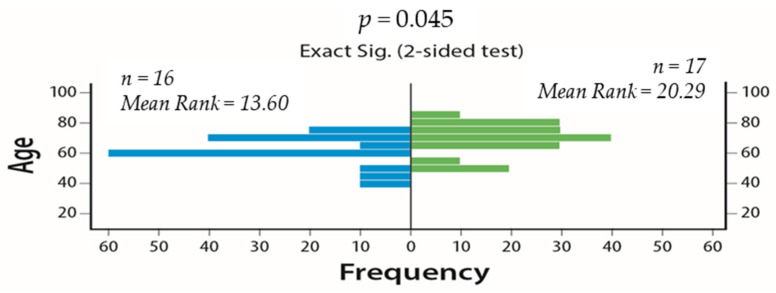
Effect of age as a continuous variable on intracranial (IC) disease control (DC) rates (Mann–Whitney U test, 95% CI). On the left side is the age of the patients that experienced intracranial disease control and on the right side is the age of those who developed intracranial progressive disease.

**Figure 3 cancers-13-01562-f003:**
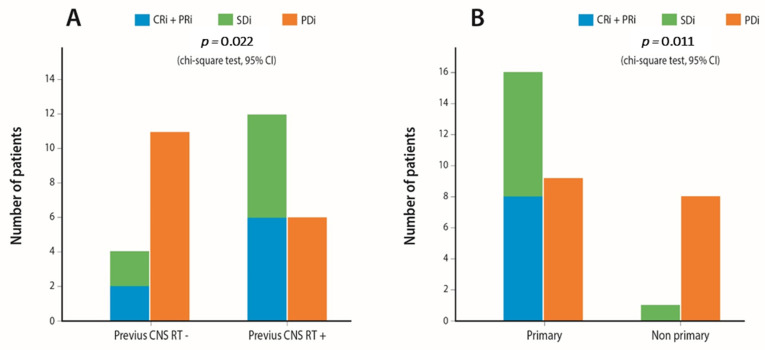
Bar plots demonstrating the effect on IC immune checkpoint inhibitor (ICI) response of (**A**). Previous IC radiation (**B**). Primary vs. non-primary brain metastases (BM) on IC DC rates.

**Figure 4 cancers-13-01562-f004:**
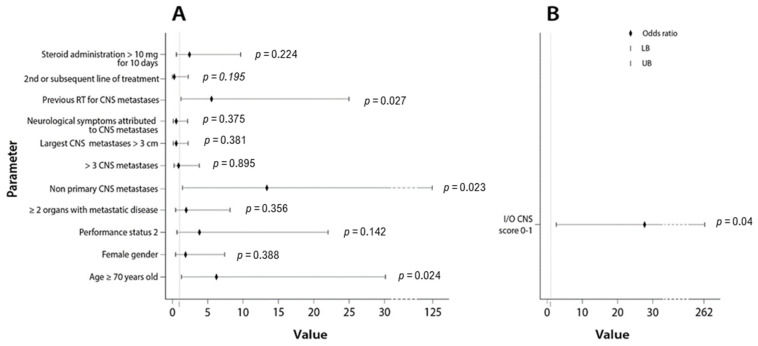
(**A**). Forest plot depicting the odds ratios of the studied parameters on the probability of developing IC PD as a response to ICI administration; (**B**). Forest plot depicting the odds ratio of the immunotherapy (I-O) central nervous system (CNS) score 0–1 on the probability of developing IC PD as a response to ICI administration.

**Figure 5 cancers-13-01562-f005:**
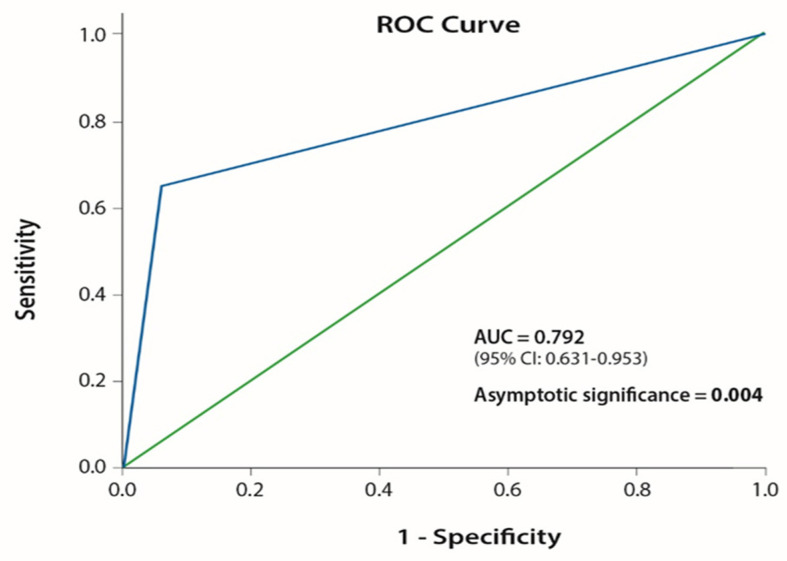
ROC curve of the I-O CNS score on the probability of development of IC PD as a result of ICI administration.

**Figure 6 cancers-13-01562-f006:**
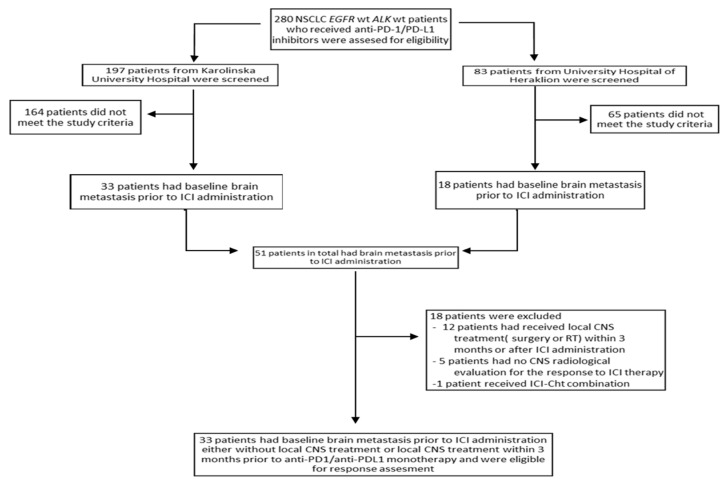
Flow chart depicting the design of our study.

**Table 1 cancers-13-01562-t001:** Patient characteristics.

Patient Characteristics		Total (*n* = 33)	95% Confidence Intervals
Age (years)	Median	69	
	Range	40–84	
Gender (*n*%)	Male	16 (48.5%)	
	Female	17 (51.5%)	
ECOG performance status (*n*%)	0–1	25 (75.8%)	
	2	8 (24.2%)	
Smoking status (*n*%)	Former or active smoker	29 (87.9%)	
	Never smoker	4 (12.1%)	
Histology (*n*%)	Adenocarcinoma	29 (87.9%)	
	Non-adenocarcinoma	4 (12.1%)	
Line of treatment of ICIs administration (*n*%)	1st line	5 (15.2%)	
	2nd or subsequent line	28 (84.8%)	
PD-L1 expression in extracranial lesions (*n*%)	<1%	2 (6.1%)	
	1% ≤ PDL1 < 50%	10 (30.3%)	
	PDL1 ≥ 50%	14 (42.4%)	
	Missing data	7 (21.2%)	
CNS metastases at diagnosis	Yes	24 (72.7%)	
	No	9 (27.3%)	
Neurological symptoms attributed to CNS metastases	Yes	18 (54.5%)	
	No	15 (45.5%)	
Radiological evaluation method of CNS metastases	MRI	23 (69.7%)	
	CT	10 (30.3%)	
Size of largest CNS metastasis (mm)	Median	17	
	Range	4–50	
Number of CNS metastases	Median	2	
	Range	1–50	
>3 CNS metastases	Yes	12 (36.4%)	
	No	21 (63.6%)	
Bone metastases (*n*%)	Yes	12 (36.4%)	
	No	21 (63.6%)	
Liver metastases (*n*%)	Yes	10 (30.3%)	
	No	23 (69.7%)	
Disease burden (*n*%)	Number of organs with metastatic disease >2	13 (39.4%)	
	Number of organs with metastatic disease ≤2	20 (60.6%)	
Steroid administration > 10 mg for > 10 days (*n*%)	Yes	16 (48.5%)	
	No	17 (51.5%)	
CNS response to previous systemic treatment	PRi or SDi	7 (21.2%)	
	PDi	21 (63.6%)	
	No previous treatment	4 (12.2%)	
	Missing data	1 (3%)	
Active CNS metastases	Yes	25 (75.8%)	
	No	8 (24.2%)	
Previous surgery for CNS metastases	Yes	4 (12.1%)	
	No	29 (87.9%)	
Previous radiation treatment for CNS metastases	WBRT	9 (27.3%)	
	SRS	9 (27.3%)	
	No RT	15 (45.5%)	
Intracranial response to ICIs*	CRi	1 (3%)	
	PRi	7 (21.2%)	
	SDi	8 (24.2%)	
	PDi	17 (51.5%)	
Intracranial response to ICIs in patients with active CNS metastases (*n* = 25)	CRi	1 (4%)	
	PRi	5 (20%)	
	SDi	2 (8%)	
	PDi	17 (68%)	
I-O CNS score	0.5–1	12 (36.4%)	
	1.5–2	21 (63.6%)	
Duration of intracranial response (months)	Median	7.53	0–18.45
	Range	0.5–22.47	
Intracranial progression (*n*%)	Yes	28 (84.8%)	
	No	5 (15.2%)	
Death (*n*%)	Yes	25 (75.8%)	
	No	8 (24.2%)	
Intracranial TTP (months)	Median	4.3	2.81–5.79
	Range	0.17–22.87	
OS (months)	Median	6.77	2.34–11.19
	Range	0.5–30.3	
Follow-up (months)	Median	6.67	2.31–11.19
	Range	0.13–22.13	

**Abbreviations**: ICIs = Immune Checkpoint Inhibitors, I-O = Immunotherapy, WBRT = Whole-Brain Radiotherapy, SRS = Stereotactic Radiosurgery. CR: Complete response, PR = Partial response, SD = Stable disease, PD = Progressive disease, TTP: Time to progression, OS: Overall Survival. * i stands for intracranial.

**Table 2 cancers-13-01562-t002:** Univariate binary regression analysis for the prediction of the probability of having intracranial disease progression as a response to ICI administration. ROC analysis of the predictive performance regarding intracranial progression.

	Univariate		ROC Analysis
Variable	OR (95% CI)	*p*-Value	AUC (95% CI)
Age ≥ 70 years old	6.19 (1.27–30.170)	0.024	0.700 (0.518–0.883)
Female gender	1.84 (0.461–7.312)	0.388	0.575 (0.378–0.773)
Performance status 2	3.82 (0.641–22.744)	0.141	0.614 (0.420–0.808)
>2 organs with metastatic disease	1.95 (0.471–8.130)	0.356	0.579 (0.382–0.776)
Non-primary CNS metastases	13.33 (1.424–124.880)	0.023	0.704 (0.523–0.885)
>3 CNS metastases	0.91 (0.220–3.758)	0.895	0.511 (0.311–0.711)
Largest CNS metastases >3 cm	0.52 (0.120–2.248)	0.381	0.577 (0.373–0.781)
Symptoms attributed to CNS metastases	0.53 (0.133–2.141)	0.375	0.577 (0.380–0.775)
Previous RT for CNS metastases	5.49 (1.210–25.000)	0.027	0.699 (0.515–0.882)
2nd or subsequent line of treatment	0.22 (0.021–2.191)	0.195	0.586 (0.390–0.783)
Steroid administration >10 mg for ≥10 days	2.38 (0.588–9.646)	0.224	0.607 (0.411–0.802)
I-O CNS score 0–1	27.50 (2.883–262.34)	0.004	0.792 (0.631–0.953)

**Table 3 cancers-13-01562-t003:** I-O CNS score point allocation system for the creation of two different predictive groups.

Parameter		Point Allocation
Age (years)	<70	0.5
	≥70	0
CNS metastases at presentation	Yes	1
	No	0
Previous CNS RT	Yes	0.5
	No	0

**Table 4 cancers-13-01562-t004:** Patient distribution according to their I-O CNS score.

Predictive Groups	I-O CNS Score
Group A (good)	1.5–2
Group B (poor)	0–1

## Data Availability

Not applicable.

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
