# Peer review of "Correlation of Clinical Parameters with Intracranial Outcome in Non-Small Cell Lung Cancer Patients with Brain Metastases Treated with Pd-1/Pd-L1 Inhibitors as Monotherapy"

_cancers, 2021, doi:10.3390/cancers13071562_

Round 1
Reviewer 1 Report
The premises for the study - prognostic factors in patients with brain metastases, treated with ICI - are interesting.
Overall, the study is appropriately conducted and fairly well written.
However, in the current form the finding from study would have very limited applicability for clinical decision making.
The authors have identified 33 patients with brain metastases from a primary cohort of 280 patients treated with ICI.
11 different prognostic markers are tested, and it appears that no effort have been made to account for potential multiple testing in this small cohort. (Bonferroni or similar).
Furthermore, the 3 adverse prognostic markers have extremely wide confidence intervals, due to the low number of cases:
BM not present at diagnosis, 9/33 pts., 9%% 1.4-124
No previous RT for CNS mets, 15/33pts, 95% CI 1.2-25
Age>=70years, 13/33pts, 95%CI 1.3-30.2.
As the authors state, there should be considered a high risk of type-2 error, and combining very uncertain parameters into a combined I-O CNS score, is unlikely to yield any additional gain.
For possible clinical applicability, I would suggest that the authors validate the findings in an independent dataset (according to REMARK guidelines or similar).
The following minor revisons are suggested:
1. I do not believe that Figures 2, and supp. figure 1 and 2 add any additional information other than the Mann-Whitney U test results
2. It is uncler when steroid use (<>10mg) administration is used. Before or during ICI treatment?
3. It is unclear how TTP is defined (Time to progressive disease? Or Time to progressive disease or death)?
4. It is unclear at which intervals the patients have scans performed. Figure 1 seems to suggest that som patients progress within 1-2 months?
5. The methods described, state that only RECIS 1.1 radiological assesment is used for response assessment, but in results it would appear, that also patients with clinical detoriation are counted as PD. This should be clearly descirbed in methods.
6. Data on systemic response are not shown, but could likely impact on general progression and overall survival
7. The authors speculate, that an abscopal effect may account for the impact of previous RT, however this is less likely given that RT was given >3 months before start of ICI, and no data are shown on the interval from RT to start of ICI.
8. Suppl. figure 1 shows N=15+16=31 pts. and Suppl. figure 2 show N=16+16=32pts.
It is unclear why there are 2 ptients missing (Table 1 does not reflect, that there would be missing data on size or number of metastases?
Author Response
√
Thank you very much for the detailed and scientifically sound review of our manuscript and your important comments towards the improvement of our research paper. Herein, we will answer to your comments:
MAJOR COMMENT: Thank your very much for this very important comment. We agree with you that the data and the prognostic score that we demonstrate in this manuscript are not meant for use in the daily clinical practice due to the small statistical sample. We have commented on that on the sixth paragraph of the discussion section of our manuscript. We conducted this study with the purpose of investigating the effect of clinically relevant parameters on the intracranial outcome of immunotherapy on patients with NSCLC. Our results demonstrate important clinical information that could potentially guide future research projects. Unfortunately, at the time being, we do not have access in a larger validation cohort but it is an ongoing project by our group. As we mentioned in our manuscript there is a high probability for type II errors when analyzing data from small cohorts. On the contrary, the probability for type I error is not dependent on sample size. Thus, we did not use Bonferroni correction because it is a conservative statistical method. By using a conservative correction test such as the Bonferroni correction there is as high risk of increasing the probability of type II error, thus missing important statistical associations on the effect of the analyzed parameters on the intracranial outcome of immunotherapy. Furthermore, we did not perform any multivariate analysis due to the small statistical sample of our cohort.
Minor comments:
- We used these figures in order to visualize the differential distributions of the studied parameters between individuals who achieved disease stabilization intracranially and those who experienced intracranial disease progression. We agree that they are not too important, if you think that they are redundant we can remove them.
- Thank you very much for this comment. We agree with you that we had not defined thoroughly the timeline of steroid administration in our cohort. Patients were categorized as having received steroids if they had received > 10 mg of prednisolone equivalent for > 10 days 15 days before immunotherapy initiation or during the duration of treatment. We have added that information in our manuscript, in the 4.2 section of the methods, in the 3rd
- Thank you very much for this comment. Intracranial Time to Progression was defined as the time interval between immunotherapy initiation until intracranial progression. Overall Survival was defined as the time duration between immunotherapy initiation until death. We have modified our manuscript in the methods section, 4.2, 3rd
- Thank you very much for this important comment. All the individuals that were classified as having intracranial disease progression had adequate CNS radiological evaluation that had confirmed the CNS disease status in comparison with their baseline scans. This applies to the individuals that had radiologically confirmed intracranial disease progression within 6 weeks after ICI initiation. We have verified that in the methods section, 4.3, 1st paragraph. Addressing your comment about the 5 patients that had progressed within 1-2 months, there were 5 patients that had experienced neurological deterioration due to their brain metastases and the radiological evaluation that was performed confirmed the intracranial disease progression in comparison with their baseline scans. We have made that clear in the 2.1 section of the results in the 1st paragraph.
- Thank you for this comment. All the individuals in our cohort that were classified as having intracranial progression they had radiological evaluation that had confirmed their intracranial disease progression in comparison with their baseline scans. We have verified that in the methods section, 4.3, 1st paragraph. We agree with you that it is not clear in the text about the classification of the 5 patients that experienced neurological deterioration within 6 weeks after ICI initiation. These 5 patients experienced IC experienced neurological deterioration that was attributed to their brain metastases earlier than 6 weeks after ICI initiation. These 5 individuals had subsequent radiological evaluation that confirmed IC disease progression in comparison with their baseline scans and they were classified as having IC disease progression. We have clarified that in the 1st paragraph of the 2.1 section of the results.
- Thank you very much for this important comment. The purpose of this study was to examine the intracranial efficacy of immunotherapy, thus we did not present data on extracranial response rates. 3 of our patients experienced intracranial disease progression with stable disease extracranially and 9 of our patients had extracranial disease progression whereas their brain metastases were stable under immunotherapy. The overall discordance rate between intracranial and extracranial disease stabilization rates was 36.3% in the whole cohort, and 24% amongst the individuals with active brain metastases which is in accordance with previous publications. The 9 patients that experienced extracranial disease progression with stable disease intracranially they all had treatment cessation due to extracranial progression. Their median intracranial TTP was 4.27 months (95% CI: 2.17-6.18) whereas their extracranial PFS was 1.33 months (95% CI: 1.24-1.43). The median intracranial TTP of these 9 patients was the same with the whole patient population (as demonstrated in table 1 of our manuscript), thus immunotherapy cessation due to extracranial progression did not affect intracranial outcomes in these individuals. In addition, due to the small number of patients further subgroup analysis is not possible. Based on the aforementioned reasons we did not present these data in our manuscript.
- We agree with you on this comment. We mentioned the abscopal effect as a well recognized phenomenon that demonstrates the potential systemic immunogenic effects of radiation therapy. We did not have the purpose to state that the effect of previous radiation therapy on brain metastases is due to abscopal effect. In order to avoid any further confusion, we have deleted this sentence.
- We agree with you on this comment. The numbers are wrong in the figures due to typo error. We have corrected that and demonstrate the supplementary figures with the correct numbers.
Reviewer 2 Report
Rounis et al. investigate in their manuscript a large group of patients with intracranial metastases from a primary non-small cell lung carcinoma. They try to identify clinical parameters that are correlated with a positive reaction of the intracranial tumours towards Pd-1/Pd-L1 inhibitor therapy. So far the only parameter associated with a good clinical efficacy of this therapy is the expression of the PD-L1 protein by tumour cells.
The authors analyse a large number of clinical parameters and find only few factors positively correlated with the efficacy of the Pd-1/Pd-L1 inhibitor therapy on the progression of intracranial tumours, preliminary the age, being younger than 70 years old, but as well having received radiotherapy.
This manuscript presents a good and scientifically founded condensation of the many clinical factors that might influence the success the success of Pd-1/Pd-L1 inhibitor therapy. It is not and tries not to describe a mechanistic explanation.
With this the manuscript can be published as it is.
Author Response
Thank you very much for the review of our manuscript.
Reviewer 3 Report
Submitted manuscript titled “Correlation of Clinical Parameters with Intracranial Outcome in Non-Small Cell Lung Cancer Patients with Brain Metastases Treated with Pd-1/Pd-L1 Inhibitors as Monotherapy”, by Rounis, et al investigated parameters for predicting response to Pd-1/Pd-L1 inhibitors in patients with non-small cell lung cancer with brain metastases. They reported that old age, prior radiation treatment and brain metastasis present at diagnosis are predictors of Pd-1/Pd-L1 inhibitors. Although the number of samples is small, this study has clinically important results.
- What does "PD-L1 expression levels of peripheral lesions" mean? In which organs was PD-L1 expression measured?
- Of the 33 samples, were there any patients with brain samples taken? If there is a brain sample, it is necessary to measure PD-L1 expression in this sample.
- For more accurate prognostic analysis, survival analysis (overall survival, time to progression) for three parameters, I-O CNS score is required.
- The parameter, “PD-L1 expression level in peripheral lesions” should also be included in the binary logistic regression analysis.
Author Response
√
Thank you very much for the detailed and scientifically sound review of our manuscript and your important comments towards the improvement of our research. Herein, we will answer to your comments:
- Thank you very much for this important comment. “PD-L1 levels from peripheral lesions” in our manuscript stands for PD-L1 expression levels in extracranial metastatic sites. In order to define that comment more clearly we have added additional information in the 4.2 section of the methods, 3rd paragraph and in table 1. As we mention in the revised manuscript we state that in all analyzed individuals PD-L1 expression levels were calculated from extracranial disease sites such as lungs, liver and bones.
- PD-L1 expression levels was available in 26 individuals in our cohort. In all of these 26 individuals PD-L1 levels were calculated in extracranial sites. We have clarified that in the 4.2 section of the methods, 3rd Unfortunately, we did not have any PD-L1 expression from a biopsy of a brain metastasis. We have mentioned that in the 4.2 section of the methods, 3rd paragraph.
- Thank you for this important comment. We had performed initially a univariate Cox regression analysis on the effect of the analyzed parameters on intracranial time to progression and overall survival (Table 1). None of the analyzed parameters affected intracranial time to progression or overall survival, probably due to small statistical sample. I-O CNS score affected intracranial time to progression at a statistical significant level but not OS. Due to the fact that none of the parameters that comprise the I-O CNS score affected intracranial time to progression or overall survival we did not include these results in our manuscript.
|
COX REGRESSION |
Intracranial TTP |
OS |
||
|
UNIVARIATE ANALYSIS |
HR (95% Confidence Intervals) |
p value |
HR (95% Confidence Intervals) |
p value |
|
Age ≥ 70 years old |
0.51 (0.24-1.10) |
0.09 |
0.49 (0.22-1.12) |
0.09 |
|
Female gender |
1.72 (0.80-3.67) |
0.16 |
1.47 (0.66-3.26) |
0.34 |
|
Performance status 2 |
2.10 (0.89-4.91) |
0.09 |
1.13 (0.45-2.84) |
0.79 |
|
>2 organs with metastatic disease |
1.25 (0.57-2.70) |
0.58 |
1.98 (0.85-4.62) |
0.11 |
|
Non-primary CNS metastases |
2.14 (0.91-5.03) |
0.08 |
1.49 (0.61-3.65) |
0.38 |
|
> 3 CNS metastases |
0.86 (0.40-1.85) |
0.70 |
0.92 (0.41-2.04) |
0.83 |
|
Largest CNS metastases > 3 cm |
0.60 (0.27-1.37) |
0.23 |
1.04 (0.45-2.41) |
0.93 |
|
Symptoms attributed to CNS metastases |
0.96 (0.45-2.04) |
0.92 |
0.98 (0.44-2.16) |
0.95 |
|
Previous RT for CNS metastases |
0.48 (0.22-1.04) |
0.06 |
0.66 (0.30-1.46) |
0.30 |
|
2nd or subsequent line of treatment |
0.78 (0.27-2.26) |
0.64 |
1.95 (0.57-6.61) |
0.28 |
|
Steroid administration > 10 mg for ≥ 10 days |
0.99 (0.47-2.09) |
0.98 |
0.72 (0.32-1.61) |
0.43 |
|
I-O CNS score 0-1 |
1.34 (1.04-1.73) |
0.022 |
1.26 (0.95-1.68) |
0.11 |
Table 1: Univariate Cox Regression Analysis on the effect of the analyzed parameters on intracranial time to progression (TTP) or overall survival (OS).
- PD-L1 status was available in 26 patients in our cohort. As demonstrated in table 1, only 2 patients out of these 26 had PD-L1 < 1%. Due to this fact, a binary regression analysis examining the effect of PD-L1 experssion on intracranial outcome was not possible. We have added this fact to the limitations of our study, in the discussion section, 7th
Round 2
Reviewer 1 Report
Thank you for sufficient answers to the questions raised in the review.
I think the article overall has improved after corrections.
It would strengthen the article, if the findings can be validated in an independent cohort. But I can accept that the clarification of this study as hypothesis generating - is sufficient.
Reviewer 3 Report
The authors revised the paper well according to my comments.